# Exploring single-path Architecture Search ranking correlations

## Abstract

Recently presented benchmarks for Neural Architecture Search (NAS) provide the results of training thousands of different architectures in a specific search space, thus enabling the fair and rapid comparison of different methods. Based on these results, we quantify the ranking correlations of single-path architecture search methods in different search space subsets and under several training variations; studying their impact on the expected search results. The experiments support the few-shot approach and Linear Transformers, provide evidence against disabling cell topology sharing during the training phase or using regularization and other common training additions in the NAS-Bench-201 search space. Additionally, we find that super-network size and path sampling strategies require further research to be understood better.

## 1 Introduction

The development and study of algorithms that automatically design neural networks, Neural Architecture Search (NAS), has become a significant influence in recent years; owed to the promise of creating better models with less human effort and in shorter time.

Whereas the first generations of algorithms required training thousands of networks in thousands of GPU hours using reinforcement learning (Zoph & Le (2016); Zoph et al. (2018)), greedy progressive optimization (Liu et al. (2018a)), regularized evolution (Real et al. (2018)) and more, the invention of weight sharing during search (Pham et al. (2018)) reduced the computation cost to few GPU hours, and thus made NAS accessible to a much wider audience.

While this also enables gradient based NAS (Liu et al. (2018b)), the necessity to compare operations against each other leads to an increased memory requirement. The issue is commonly alleviated by training a small search network consisting of cells with a shared topology, later scaling the resulting architecture up by adding more cells and increasing the number of channels. Although the stand-alone network is often trained from scratch, reusing the search network weights can increase both training speed and final accuracy (Yan et al. (2019); Hu et al. (2020)). More recent gradient based methods require to have only one path in memory (Dong & Yang (2019); Cai et al. (2019); Hu et al. (2020)) and can even be applied directly to huge data sets.

However, the aforementioned weight sharing methods only yield a single result, require manually fine-tuning the loss function when there are multiple objectives, and can not guarantee results within constraints (e.g. latency, FLOPs). The single-path one-shot approach seeks to combine the best of both worlds, requiring only one additional step in the search phase (Guo et al. (2020)): Firstly a full sized weight-sharing model (super-network) is fully trained by randomly choosing one of the available operations at each layer in every training step. Then, as specific architectures can be evaluated by choosing the model's operations accordingly, a hyper-parameter optimization method can be used to find combinations of operations maximizing the super-network accuracy. If the rankings of the architectures by their respective super-network accuracy and by their stand-alone model retraining results are consistent, the quality of the discovered candidates is high.

However, since the single-path method's search spaces are often gigantic and the network training costly (see e.g. Guo et al. (2020); Chu et al. (2019b;a)), a study of the ranking correlation is usually limited to a handful of architectures. In this work we study the single-path one-shot super-network predictions and ranking correlation throughout an entire search space, as all stand-alone model re-

sults are known in advance. This enables us to quantify the effects of several super-network training variations and search space subsets, to gain further insights on the popular single-path one-shot method itself.

We briefly list the closest related work in Section 2 and introduce the measurement metric, benchmark dataset, super-network training and experiment design in Section 3. We then systematically evaluate several variations in the single-path one-shot approach with a novel method, computing the ranking correlation of the trained super-networks with the ground-truth top-N best architectures. Experiments on search space subsets in Section 4.1 once again demonstrate that the ranking is more difficult as the search space increases in size, and that the operations that make the ranking especially hard are Zero and Pool. Section 4.2 evaluates Linear Transformers (Chu et al. (2019a)), which we find to perform very well in specific search space subsets, and otherwise even harmful. Furthermore, some commonly used training variations such as learning rate warmup, gradient clipping, data augmentation and regularization are evaluated in Section 4.3, where we find that none of these provides a measurable improvement. We further test disabling cell topology sharing only during training time and find that training the network in the same way as evaluating it is more effective. We finally list some grains of salt in Section 5 and conclude the paper with Section 6.

## 2 RELATED WORK

A high quality architecture ranking prediction is the foundation of any NAS algorithm. In this paper we explore the effects of several super-network training variations on the ranking prediction of the aforementioned single-path one-shot approach (Guo et al. (2020)). Recent efforts have shown improvements by strictly fair operation sampling in the super-network training phase (Chu et al. (2019b)) and adding a linear $1\times1$ convolution to skip connections, improving training stability (Chu et al. (2019a)). Other works divide the search space, exploring multiple models with different operation-subsets (Zhao et al. (2020)), or one model with several smaller blocks that use a trained teacher as a guiding signal (Li et al. (2020b)).

Due to the often gigantic search spaces and the inherent randomness of network training and hyperparameter optimization algorithms, the reproducibility of NAS methods has become a major concern. NAS Benchmarks attempt to alleviate this issue by providing statistics (e.g. validation loss, accuracy and latency) of several thousand different networks on multiple data sets (Ying et al. (2019); Dong & Yang (2020)), providing the ground-truth training results that we use for our evaluation.

## 3 METHOD

### 3.1 METRIC

As we correlate the super-network accuracy prediction and the benchmark results, but are only interested in a correct ranking, we need a ranking correlation metric. We choose Kendall's Tau ($\tau$, KT), a commonly used ranking metric (Sciuto et al. (2019); Chu et al. (2019b)) that counts how often all pairs of observations $(x_i, y_i)$ and $(x_j, y_j)$

1. are concordant, agreeing on a sorting order
   ($x_i < x_j$ and $y_i < y_j$; or $x_i > x_j$ and $y_i > y_j$)

2. are discordant, disagreeing on a sorting order
   ($x_i < x_j$ and $y_i > y_j$; or $x_i > x_j$ and $y_i < y_j$)

3. are neither

and is then calculated by their difference and normalized by the number of possible different pairs.

$$\tau = \frac{(num\ concordant) - (num\ discordant)}{\binom{n}{2}}$$

$\tau$ ranges from -1 in perfect disagreement to +1 in perfect agreement, and is around zero for independent $X$ and $Y$.

A small selection of experiments that use additional metrics can be found in Appendix D.

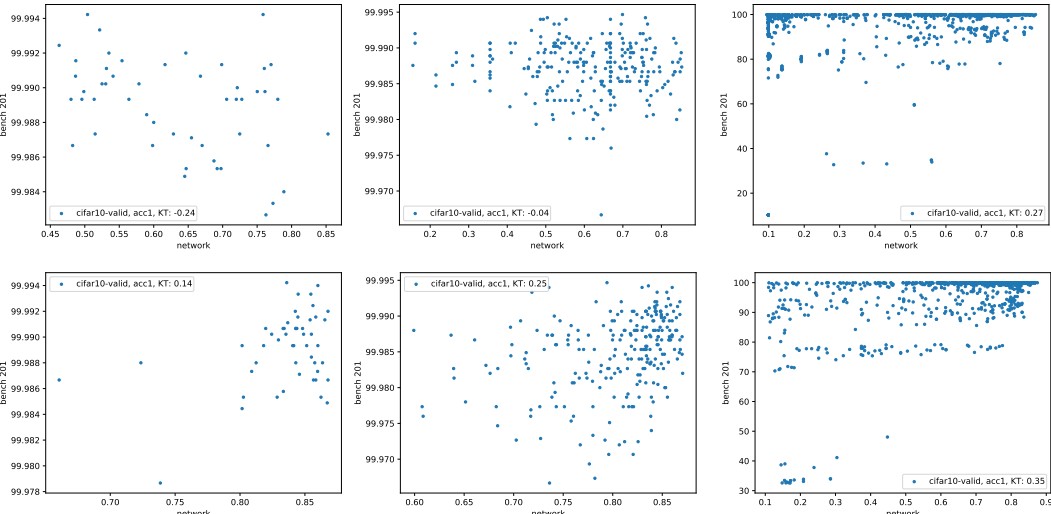

Figure 2: Accuracy predictions of a small super-network (x axis) and Bench201 results (y axis, in percent) for the top-50, top-250 (left and center columns) and 1000 randomly sampled (right column) benchmark networks on the cifar10-valid data set, ordered by accuracy. While all five operations are available in the top row, Zero is masked in the bottom row.

## 3.2 NAS-BENCH-201

NAS-Bench-201 (Dong & Yang (2020)) is a tabular benchmark, which contains training and evaluation statistics of 15625 different architectures on the common vision data sets CIFAR10, CIFAR100 (Krizhevsky et al. (2009)) and a reduced variant of ImageNet (Deng et al. (2009)). The models differ in the design of the cell, a building block that is stacked several times to create a network. Within the cell, as visualized in Figure 1, at six specific positions (orange edges), one of five operations (Zero, Skip, 1×1 Convolution, 3×3 Convolution, 3×3 Average Pooling) is chosen ($5^6 = 15625$). The inputs of each node, such as the cell output (rightmost node) are averaged.

As we are only interested in the final accuracy of each architecture, we average the benchmark test results over all seeds and the last three epochs of training. As the models' rankings are quite consistent across all data sets (Dong & Yang (2020)), we focus on the CIFAR-10-Valid accuracy. Further results are provided in the supplementary material, see Appendix B.

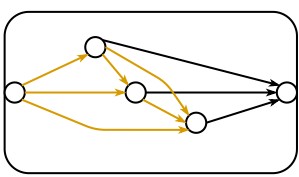

Figure 1: A NAS-Bench-201 cell with operations on the orange edges

Since discrepancies of model rankings for the top performing architectures became apparent (Dong & Yang (2020)), we measure the accuracy of the trained super-networks according to the top-N (10, 25, 50, 150, 250, 500) benchmark architectures, as well as up to 1000 randomly sampled ones. If a reduced search space (due to masking operations, $3^6 = 729$) contains fewer than 1000 different topologies, it is fully evaluated.

## 3.3 TRAINING

In our experiments we train various NAS-Bench-201 networks. Small variants have 2 cells per stage (total of 8 cells, with 3 stages and 2 fixed cells for spatial reduction) and 32 channels in the first cell, which is roughly similar to common topology sharing methods. Medium sized networks have 4 cells per stage and start with 64 channels.

All models were subject to the same training schedule. We used CIFAR10 as training set (Krizhevsky et al. (2009)), of which we withheld 5000 images for validation. The batch size is 256, we used SGD with momentum of 0.9 and learning rate of 0.025, which was cosine annealed to 1e-5 over 250 epochs. All results are averaged over five independent runs with different seeds. Further details are listed in Appendix A.

### 3.4 EXPERIMENT DESIGN

All of the following experiments are structured the same way: The top-N network architectures (ordered by top1 accuracy, measured in NAS-Bench-201) are selected, and an over-complete super-network predicts their respective accuracy values, as seen in Figure 2. If an operation is not available to the super-network, the top-N networks are also taken from the bench results without that operation.

Variations to the search space and the super-network (structure or training process) affect the ranking correlation $\tau$ between the bench results and the super-network predictions. In the case of Figure 2, removing the Zero operation from the search space improves $\tau$.

To make the figures more compact, the exact benchmark and prediction values are ignored in the further figures, only average prediction accuracy and $\tau$ depending on N will be shown (see Appendix B for further detailed figures), as seen in e.g. Figure 4. We also add the additional metric $\tau_a$ which describes the ranking correlation of the average prediction accuracy depending on N. More formally, $\tau_a$ is computed as described in Section 3.1 on the series of measurements $[(10, A_{10}), (25, A_{25}), (50, A_{50}), ...]$ where $A_N$ is the accuracy of super-network $M$ with topology $T_i$ and weights $\theta_s$ on the validation data $D_{valid}$, averaged over the top-N topologies and multiple seeds.

$$A_N = \sum_{s \in seeds} \frac{1}{|seeds|} \sum_{i=1}^{N} \frac{1}{N} Acc(M, \theta_s, T_i, D_{valid})$$

As we increase N (10, 25, 50, ...) $A_N$ should monotonically decrease (e.g. 0.7, 0.65, 0.6, ...), so that $\tau_a = -1$ is the case where the super-network estimates match the bench results best.

## 4 EXPERIMENTS

### 4.1 SEARCH SPACE SUBSETS

The search space itself plays a significant role for NAS algorithms, not only due to its size or the availability of good models. By preventing specific operations from being used (masking) during training, validation and in the benchmark results, we can compare different subsets of the search space.

A visual example of the importance can be found in Figure 2, where for an increasing number of top-N benchmark networks (columns), the ranking correlation to the super-network predictions is steadily improving. While there is a large number of networks wrongly predicted as useless (10% accuracy, right column) in the top row, masking the Zero operation (bottom row) significantly reduces this portion and thus improves the ranking correlation $\tau$ (KT).

To get a deeper understanding of the search space subsets, we take a closer look at the NAS-Bench-201 rankings, sorted by their top 1 accuracy. Specifically, in Figure 3 we count how many of the NAS-Bench top-N networks use each available operation and how often it is used, across all benchmark results and the five largest subsets (each operation is masked once).

The 3×3 Convolution is arguably the most important operation. Unless it is masked, every single top-500 Bench network makes use of it. On average, it even makes up roughly half of the operation choices, in every other subset. This is hardly surprising, since it adds significantly more capacity to the network than the 1×1 Convolution and especially Zero, Skip, or Pool.

The order of operation importance, as implied in their usage, continues with Skip, 1×1 Convolution, Zero, and finally Average Pooling. The wide usage of Skip operations was to be expected, as they are known to make deep networks more easily trainable (He et al. (2016)), however they are not present in every network. Perhaps the most surprising is the low importance of Average Pooling, even lower than Zero. It appears that all the benefits of Pool are already covered by the 3×3 Convolution, so that using the unnecessary operation now decreases the network accuracy.

Two subsets behave notably different than the full search space. Firstly, in the absence of skip connections (top right), it appears that Average Pooling is used as a substitute. And secondly, in the

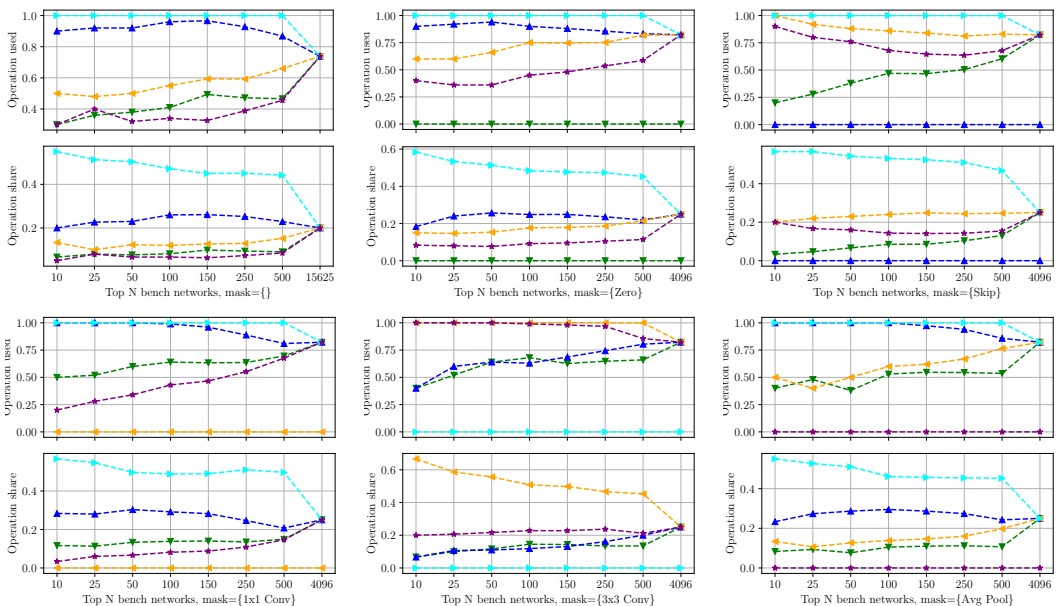

Figure 3: Counting how many top-N Bench architectures use an operation at any position in the architecture (top plot in each of the two rows and three columns) and their total usage (each bottom plot) over the top-N networks and all possible architectures (x axis). The operations are Zero (green), Skip (blue), 1×1 Conv (orange), 3×3 Conv (cyan) and 3×3 Avg Pool (purple). One operation is masked in each of the six pairs, except for the top left one. As an example, if 3 of the top-10 networks use at least one Pool operation, together a total of 4 Pool operations, and have all 5 operations available, the usage is $3/10 = 0.3$ while the share is $4/(5 \cdot 10) = 0.08$.

absence of the 3×3 Convolution (bottom center), Average Poolings and especially 1×1 Convolutions have to make up for the missing capacity and spatial operations.

We now train single-path super-networks in several search space subsets and visualize the results in Figure 4. Ideally, the super-network validation accuracy is highest for the top Bench networks, enabling NAS methods to reliably find them, and the ranking correlation $\tau$ within the top-N bench networks is always significantly greater than zero, thus increasing the expected quality of the selected architecture. Neither is the common case. The baseline for small networks (top left, red) has the same averaged prediction accuracy for the top 10 as for the top 500 networks, resulting in $\tau_a \approx 0$, where the predicted accuracy and the bench accuracy have no statistical correlation.

However, in some search space subsets, the single-path method works significantly better. By masking out each operation individually (Figure 4, left column), we find the most harmful operations to be Zero (green, top left, $\tau_a=-1$) and Pool (purple, bottom left, $\tau_a=-1$), which are also the least important ones according to Figure 3. Masking the Convolutions, thus increasing the relative amount of unparameterized operations, is harmful.

Masking Skip (blue, left) is the most harmful to $\tau_a$ (=1). As seen in Figure 4, the top-N networks have a worse average predicted accuracy than the top-M (for N < M) networks, and sometimes even below the random sample, which is terrible. Interestingly $\tau$ may improve within the predictions for the top-N architectures.

We further mask a second operation in addition to Zero (center column) and Pool (right column) in the remaining columns of Figure 4. On small networks, the masking combination of Zero+Pool and arguably Zero+Skip perform even better, while masking Pool in almost any combination is harmful.

It is quite obvious that medium sized super-networks require additional care. The super-networks in several search space subsets fail to generalize at all, even though they learn the training set. In the other spaces, they still behave differently. This may be beneficial, such as in the baseline (left, red), but is more often harmful. Even worse, the averaged predicted accuracy of top-N networks in several subsets is lower than that of a random subset of networks, despite the often improved $\tau$

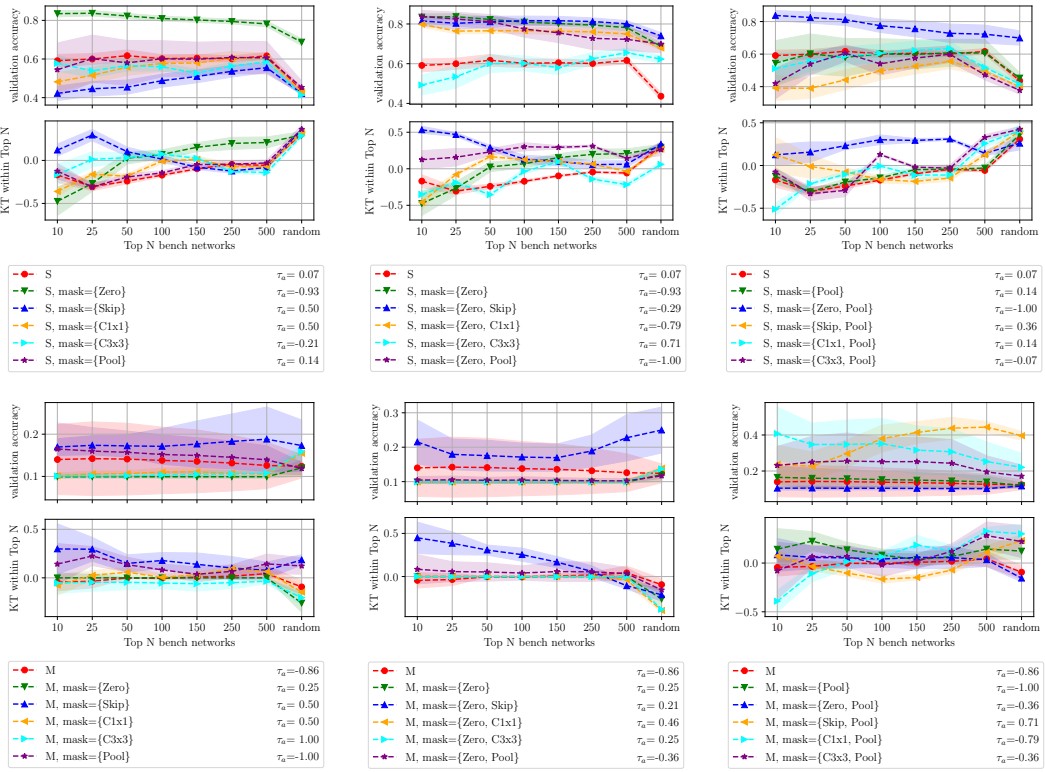

Figure 4: Measuring ranking correlations and average accuracy (y axis) of the top N bench networks (x axis) in different search spaces (color), plotting the mean and std of five runs. We used small sized (S) super-networks in the top row and medium (M) sized ones in the bottom row.

values. Finally, as seen in Figure 4, the standard deviation over the super-networks that do generalize is notably greater than for their small sized counterparts.

## 4.2 LINEAR TRANSFORMERS

Many common search spaces for single-path methods contain exclusively Convolution operations (or blocks of such). Adding a Skip operation is useful in theory, enabling the discovery of smaller sized networks, but was also found to impact the stability of the super-network training. After all, in a sequential super-network, the operations at any layer may directly receive the output of any previous layer due to the variable size. However, replacing the Skip operation with a linear $1 \times 1$ Convolution (Linear Transformer, LT) during the search phase was found to stabilize the training (Chu et al. (2019a)). All Linear Transformers are removed after the search, resulting in a standalone network with the same capacity.

Figure 5 visualizes the results of super-networks that have Linear Transformers added to their Skip or Skip+Pool operations. We also mask Zero (center) and Zero+Pool (right) to observe the super-network in search spaces with fewer operations that are neither Convolution nor Skip. It is noteworthy that in the absence of the Pool operation (right), both variations of the standard super-network are in fact equal. This is also apparent in the plot, although the randomness of non-deterministic training can still be seen.

We find it an interesting observation that, unless the search space contains exclusively Skip or Convolutions (the context in which the Transformers have been proposed), the Transformers are always harmful to the ranking correlation $\tau_a$. The super-networks seem to overestimate the benefit of Skip and Pool operations, which, again, often causes the best networks to be estimated as below average. However, in a fitting search space, even the medium sized super-networks generalize very successfully, have a very low standard deviation and reliably improve $\tau_a$.

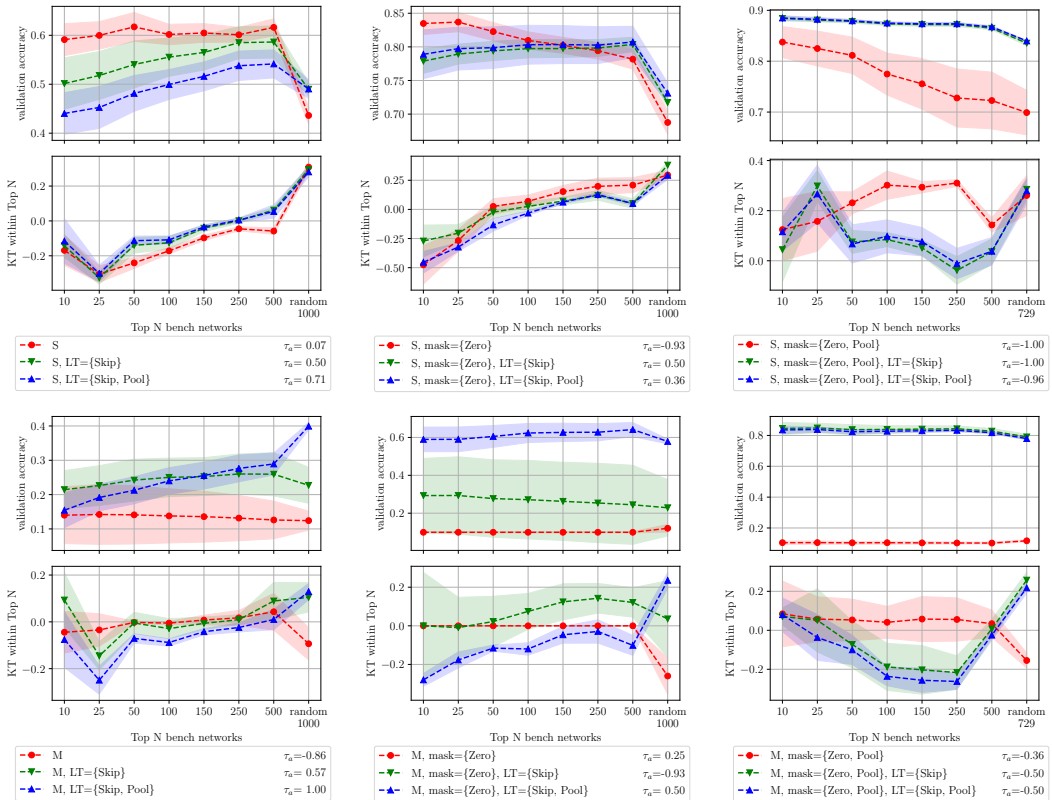

Figure 5: Adding Linear Transformers (LT) to Skip and Skip+Pool (color).

Finally, are Linear Transformers for a Pool operation useful? Since the search spaces that suit single-path networks best do not contain any Pool operation, they are generally not necessary. When a Pool operation is present, as in Figure 5 left and center-left, we find no empirical evidence that they are beneficial. Although the additional transformers seem to stabilize training, as seen by the lower standard deviation, they also worsen the $\tau_a$ problem.

## 4.3 SHARING, SAMPLING, WARM-UP, AND REGULARIZATION

Finally, we group four further variations to super-network training in Figure 6 and compare them with the baseline (red).

**Topology sharing** (green): As sharing cell topologies does not impact the resource costs of single-path training, it is generally not used. However, they are shared in the NAS-Bench-201 case, raising the question whether sharing should already be enforced during the super-net training (our default case), or only for the evaluation.

As seen in Figure 6, disabling the sharing during training for small super-networks (top row) is generally not beneficial over the baseline, as $\tau_a$ is generally worse and $\tau$ almost the same. However, it enables the medium sized networks in multiple spaces to make any useful predictions at all.

**Uniform sampling** (blue): Additionally, our default baseline strategy of randomly selecting the paths during training is strictly fair, so that every $|O|$ steps, every operation $o \in O$ is sampled exactly once; and compare this with the alternative of uniform random sampling.

Interestingly, the absolute validation accuracy value is increased by uniform sampling. However, this is not relevant, as only the correct ranking matters. We find that, on small super-networks, as measured by $\tau_a$, the strictly fair baseline performs equal or better than the uniform random sampling strategy. Additionally, we see a trend of $\tau$ being slightly in favor of strictly fair randomness, at almost every data point. However, once again, a seemingly inferior method variation enables training the medium sized super-networks to make above-chance predictions on the validation set. We hypothesize this to be a downside of the strictly fair weight update schedule, in which an update is

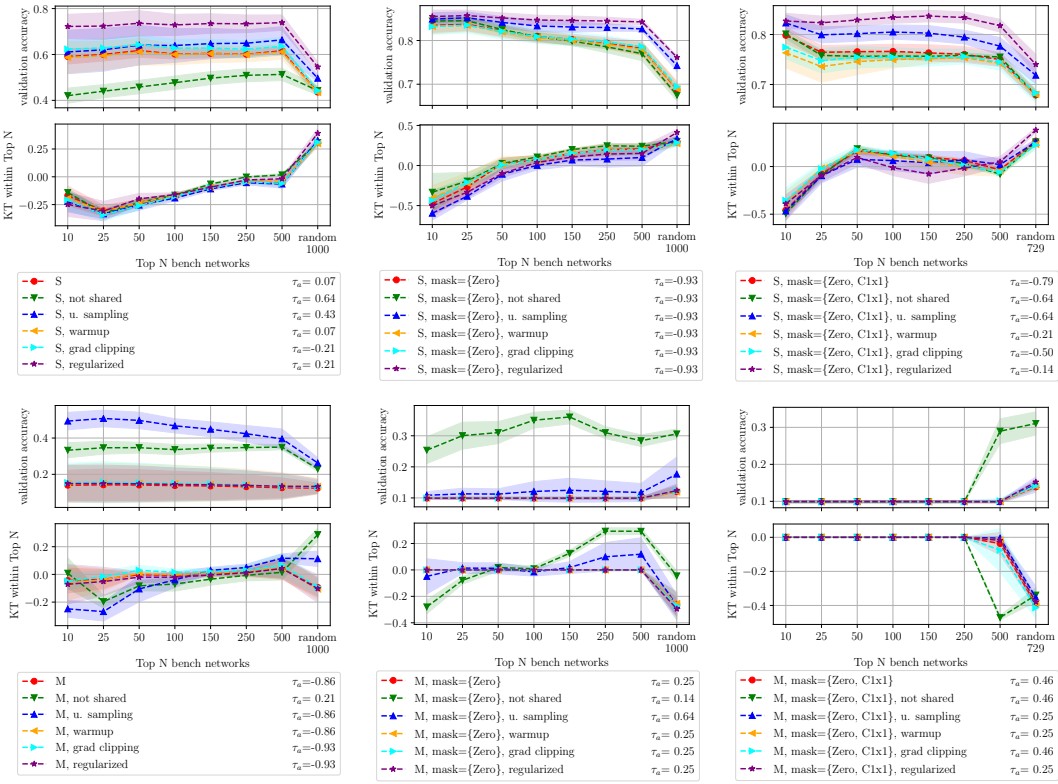

Figure 6: Further variations to the super-network training (color).

only performed every $|O|$ steps (number of operations) over the accumulated gradients, including for the weights of stem and output layer, which may result in destructively large steps.

**Learning rate warm-up** (yellow) and **gradient clipping** (cyan): To see whether simple learning rate tweaking already solves the aforementioned issue, we add a warm-up phase, linearly increasing the learning rate over 5 epochs to the default starting value of 0.025. It does not. In fact, the effects are detrimental in some search space subsets. On the other hand, clipping the gradients so that their L2 norm is in $[-5, 5]$, as common practice in e.g. DARTS (Liu et al. (2018b)), has no notable effect.

Further training variations to solve the issue with fairly little effort may be excluding the last layer and stem weights from the $|O|$-step update schedule (but losing strict fairness) or lowering their learning rate. However, a much closer look at it seems preferable, to ascertain the root cause and study its implications in greater detail.

**Regularization** (purple): Finally, the super-network is only minimally regularized by default (only input shifting, horizontal flipping, and normalizing on the data), so we add the CIFAR-10 AutoAugment augmentation policies (Cubuk et al. (2018)) and label smoothing of $\alpha = 0.1$.

Interestingly, this is also detrimental. As seen in the top left and center-right plots, $\tau_a$ decreases below the baseline values, as top ranking architectures are underestimated. The effect is worse with relatively more unparameterized operations available, indicating that the topology estimation is biased in favor of the regularized Convolutions. The effect on medium sized super-networks can not be properly measured, as none of these super-networks should be used to rank architectures.

## 5 GRAINS OF SALT

As in any empirical study, some grains of salt remain. First and foremost, the limited sample sizes in our experiments and the benchmark are a typical concern.

We find it disappointing that, aside from limited search spaces, no experiment displayed high values of $\tau$, even though the top-N network groups may be sorted correctly ($\tau_a \approx -1$). This indicates that a number of good networks are always wrongly estimated, quite likely due to some of the available operations, or that the single-path one-shot approach is simply not suitable for the given search space or network architecture. While the accuracy difference between the best and 10th-best Bench networks is only roughly $0.013\%$, masking Skip at least increases that difference to roughly $0.027\%$, possibly also making a correct ranking easier. Considering these marginal differences, it is very likely that the Bench baseline is also not perfectly correct.

Next, existing NAS benchmarks may be too small or biased. Other research has discovered surprisingly simple methods that achieve state of the art results (e.g. Mellor et al. (2020); White et al. (2020)), but suffer from an often significantly reduced performance in larger search spaces. This is hardly surprising, considering that the best architectures consist of mostly $3\times3$ Convolutions (see Figure 3).

And finally, our experiments use single-path one-shot methods, which are commonly employed in search spaces of only Convolution and Skip operations. They are our approach of choice due to their current popularity in the NAS field and the comparably cheap evaluation of many network topologies, which also enables us to study more variations of the baseline method.

## 6 CONCLUSIONS

Some search space subsets are easier to rank. In this specific case, the removal of the Zero and Pool operations keeps the majority of the top-N networks while also improves how well a single-path network can rank them.

Linear Transformers are useful when there are no other operations besides Convolutions or Skip, and enable medium sized super-networks to be used at all. However they introduce systematic ranking problems in other search space subsets, limiting their general use. We find no evidence that Pool operations with transformers are beneficial.

Disabling cell topology sharing during the super-network training decreases the ranking correlation $\tau_a$, the network should be trained the same way as it is evaluated.

Strictly fair randomness is generally advantageous, but requires further research to be understood better. Especially several medium sized super-networks were unable to generalize, in contrast to those trained with uniform sampling, which we believe to be due to the weight updates of the last layer and the stem. Simply adding learning rate warm-up or gradient clipping is insufficient to fix this issue.

Strong regularization during the super-network training was found to be detrimental. This is most likely an issue of the regularization only benefitting Convolutions, biasing the topology estimation, and may not be a problem in entirely different search spaces that use fewer to no unparameterized operations.

Whether an increased super-network size is helpful is tricky to evaluate. In few search spaces, e.g. as seen in Figure 4, the increased network size improved $\tau_a$, however the generally low validation accuracy (usually $< 20\%$, on 10 classes) and its huge variance make them too unreliable. Even worse, the super-networks may fail to generalize at all depending on the search space. In specific cases this can be alleviated with Linear Transformers, and possibly through a better understanding of path sampling. We present additional Figures in Appendix D.

Due to the limited space, we have only shown the results for the CIFAR-10-valid Bench accuracy values. However, we provide all Tensorboard (Abadi et al. (2016)) files and the code to parse and generate plots in the supplementary material.

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

# A    TRAINING SETUP

## A.1    ENVIRONMENT

Each super-network was trained and evaluated on a single Nvidia GTX 1080 Ti GPU, using driver version 440.64, CUDA version 10.0.130 and CuDNN version 7605 in our Slurm cluster. The code is run in a Singularity container using Ubuntu 18.04 with Python 3.6.9. We used PyTorch in version 1.5.1 and nas-bench in version 1.3, further details can be found in the provided *sysinfo.txt*.

## A.2    NETWORK

Aside from the deliberate variations in super-network training and the seeds, all of them were trained and evaluated in the same way, as listed in Table 1. Unless a detail is mentioned there, we are confident of not using it (e.g. we use no regularization or gradient clipping by default). The full list of arguments of each training job can be found in the respective *log_task.txt*, see Section B.2.

| | |
|---|---|
| seeds | $\{0, 1, 2, 3, 4\}$ |
| CuDNN | used, not deterministic |
| weight init. | PyTorch default |
| Data | CIFAR-10 training set |
| - for validation: | same 5000 random images for all |
| - for training: | remaining 45000 images, shuffled |
| batch size | 256 |
| Augmentation | like DARTS: |
| - shifting: | 4 pixel |
| - flipping: | horizontally |
| - normalizing: | yes |
| for validation: | only normalizing |
| Optimizer | SGD: |
| - initial LR | 0.025 |
| - momentum | 0.9 |
| - weight decay | 0.0003 |
|   - excluding BN | yes |
| Scheduler | Cosine decay: |
| - final LR | 0.00005 |
| - epochs | 250 |

Table 1: default super-network training setup

# B    PROVIDED DATA AND CODE

Please see the supplementary material for the following data and code. Due to the amount of Tensorboard files, a 7zip compression is necessary to be below the allowed 100MB limit.

## B.1    BENCH201

Since the original NAS-Bench-201 contains far more information than we need for the evaluation and requires impractically many resources (25+ GB RAM), we have a reduced version that averages results over seeds and contains only the required stats.

The data file (*nasbench201_1.1_mini.pt*) and the code to use it (*code/bench.py*) are provided.

## B.2    RUN DATA

The relevant logs and Tensorboard files of every run (slurm job) are provided in the *run_data* folder, grouped by experiments. The *code/parse_runs.py* script is used to extract desired metrics from the

these files and average them across the jobs that used different seeds. Running the script generates a text output (csv format) that is used in *plots.py*.

### B.3 PLOTTING

Running *plots.py* will use the previously generated csv text, containing the mean, standard deviation, etc. and generate the plots from the paper.

## C FURTHER FIGURES

Due to space limitations, we could not add further plots to the paper. The remaining ones for the evaluation on Cifar10-valid are found below.

If you are interested in evaluation results on the other data sets, please take a look at Appendix B.

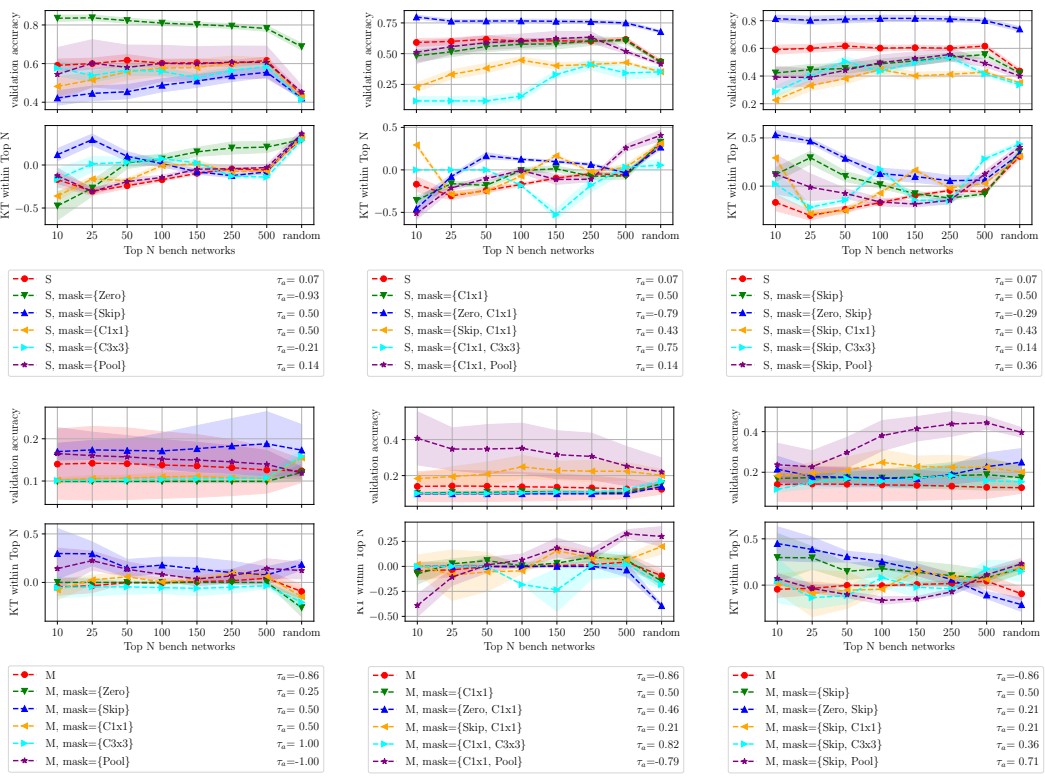

Figure 7: See Figure 4, the left column is kept the same, adding masking combinations with the 1×1 Convolution (center column) and Skip (right column)

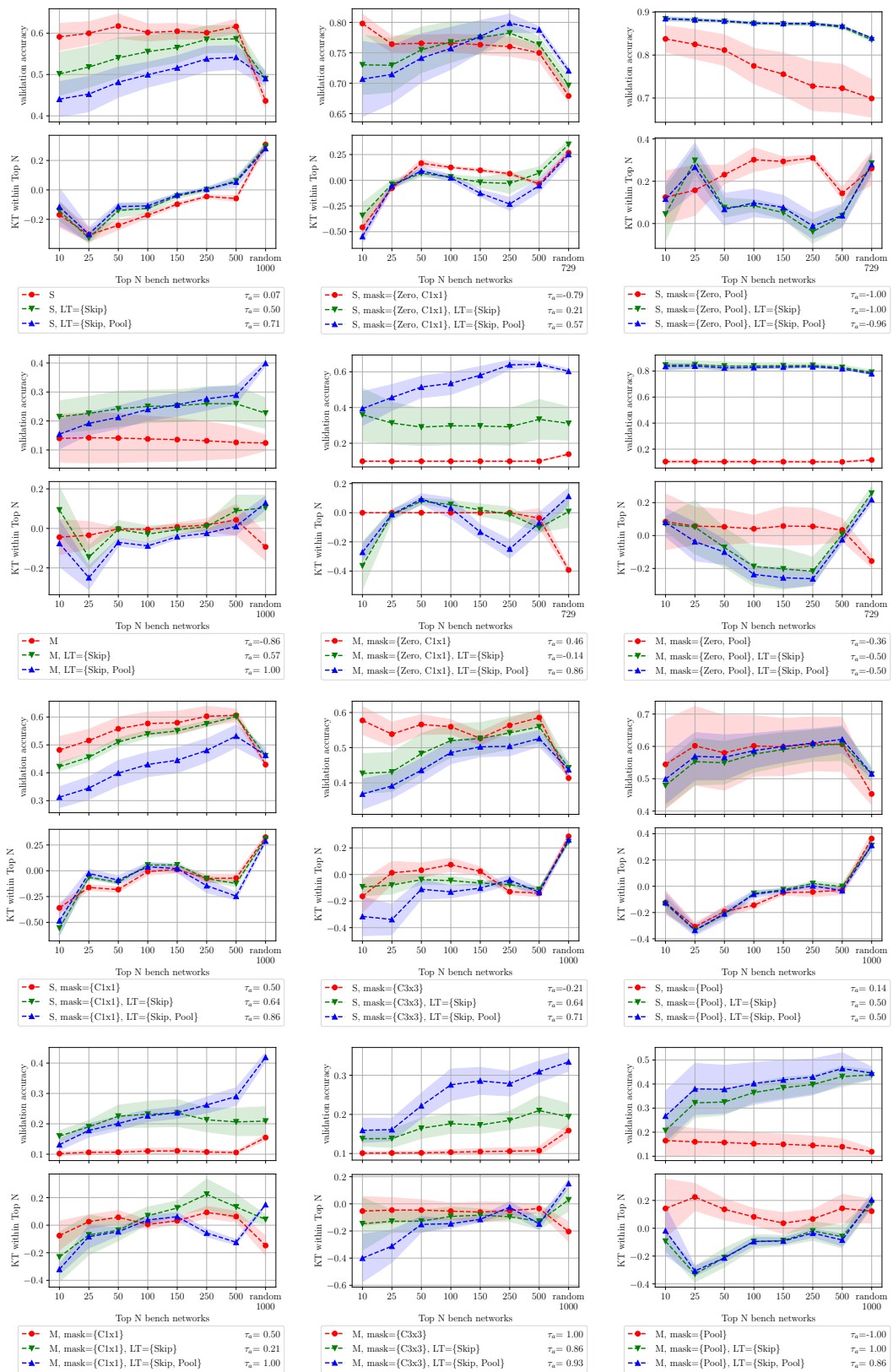

Figure 8: See Figure 5, further search space subsets with Linear Transformers.

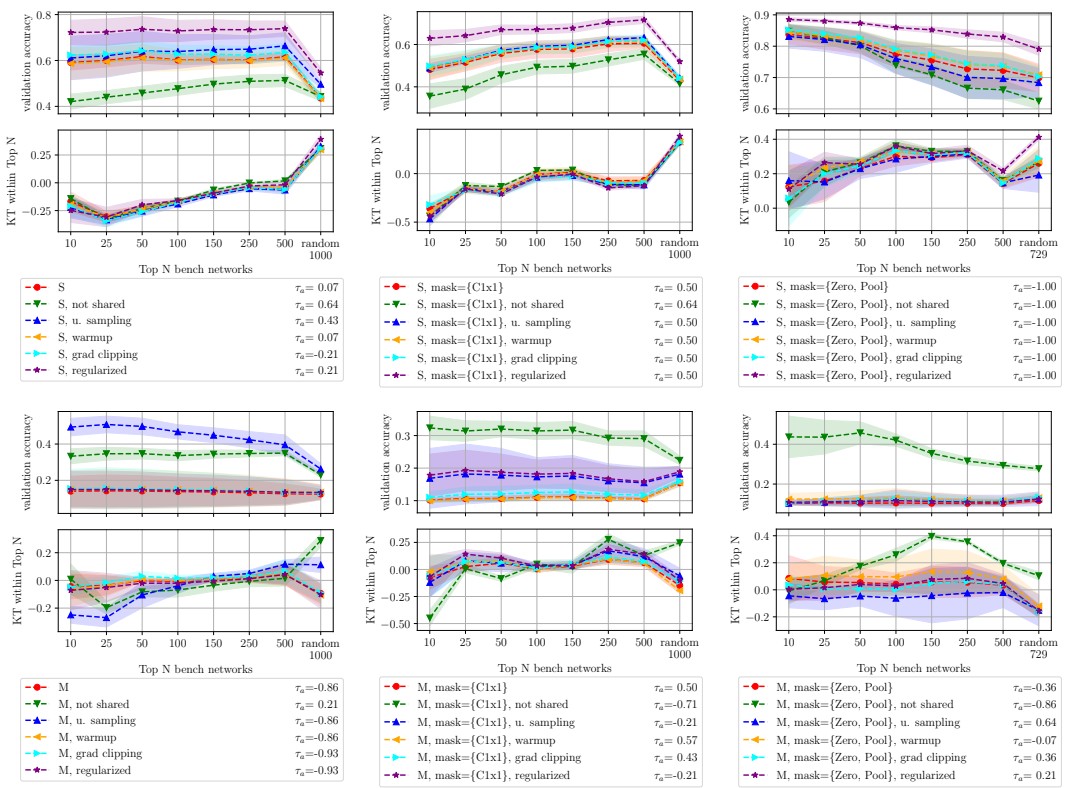

Figure 9: See Figure 9, the left column is kept the same, adding masking combinations with the 1×1 Convolution (center column) and Zero+Pool (right column)

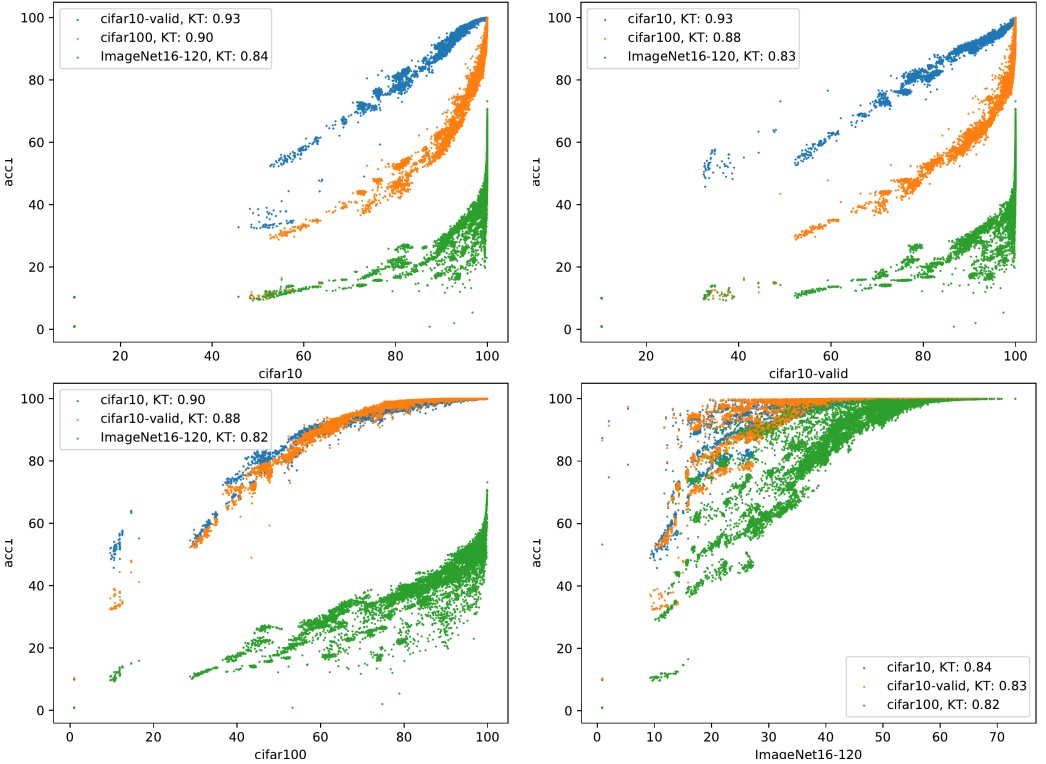

Figure 10: The accuracy values (x axis) of all architectures on the NAS-Bench-201 data sets and their respective accuracy on the other three data sets (color, y axis).

# D ADDITIONAL METRICS AND WIDE-CHANNEL SUPER-NETWORKS

In addition to Kendall's Tau, we now also provide the Spearman Correlation Coefficient (SCC) and the Pearson Correlation Coefficient (PCC) (Li et al. (2020a)) for a selection of the experiments, and additional experiments on small but wide super-networks, starting with 96 (instead of 32) channels.

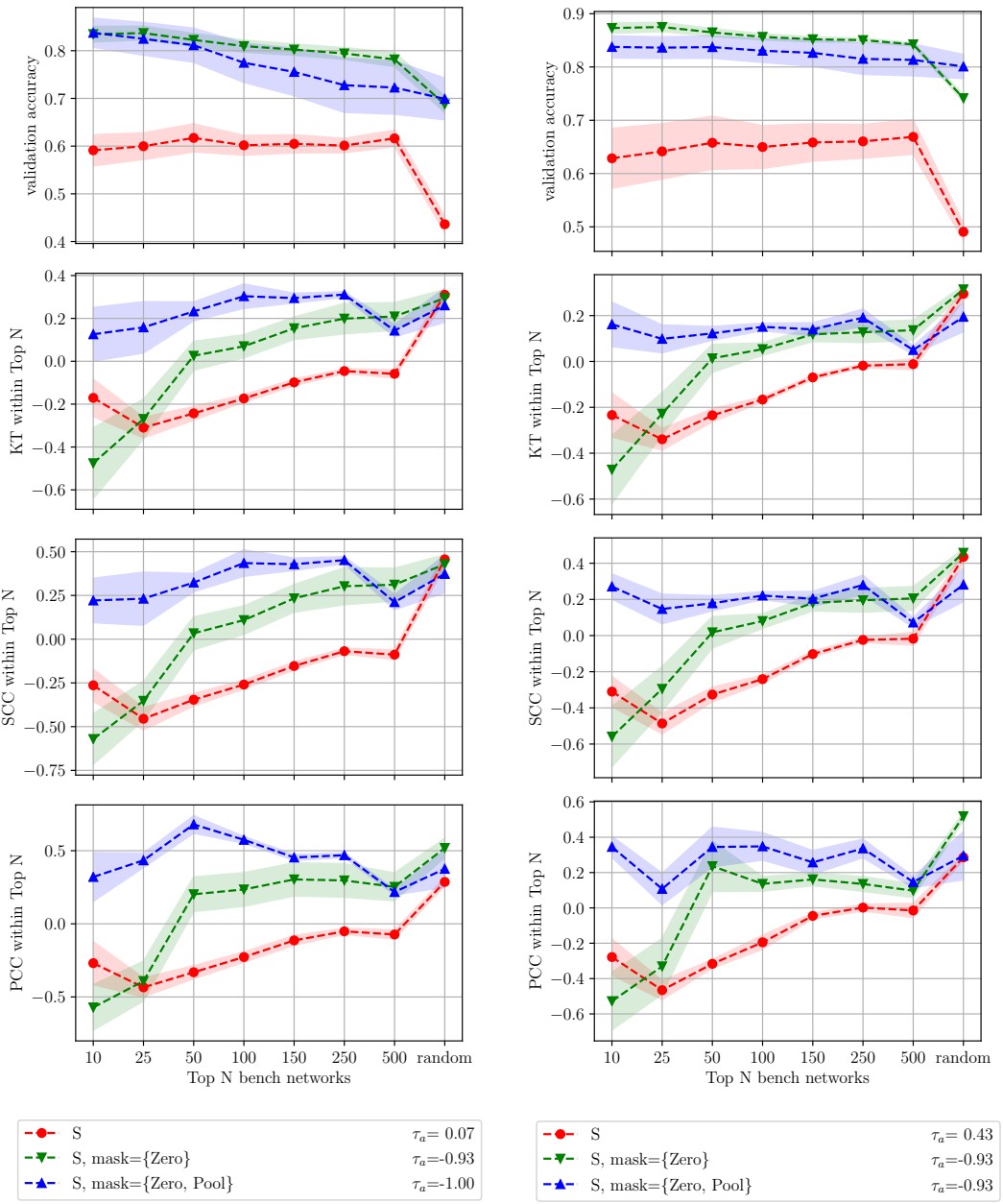

Figure 11: Additional metrics for small super-networks that start with 32 (left) or 96 channels (right).

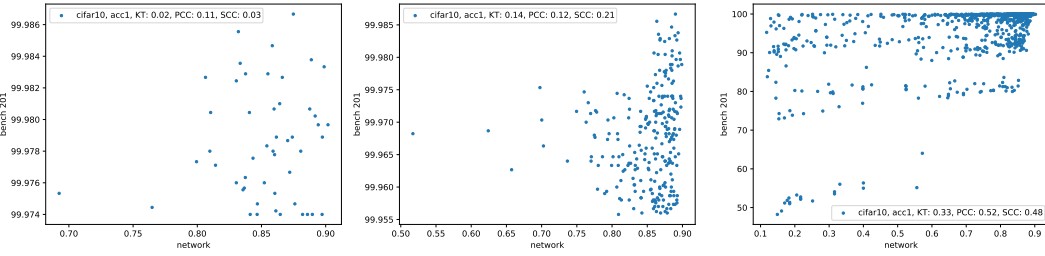

Figure 12: As Figure 11, adding linear transformers.

Figure 13: Visualizing some metrics, similar to Figure 2, of the small and wide super-networks (Figure 12, left, green)

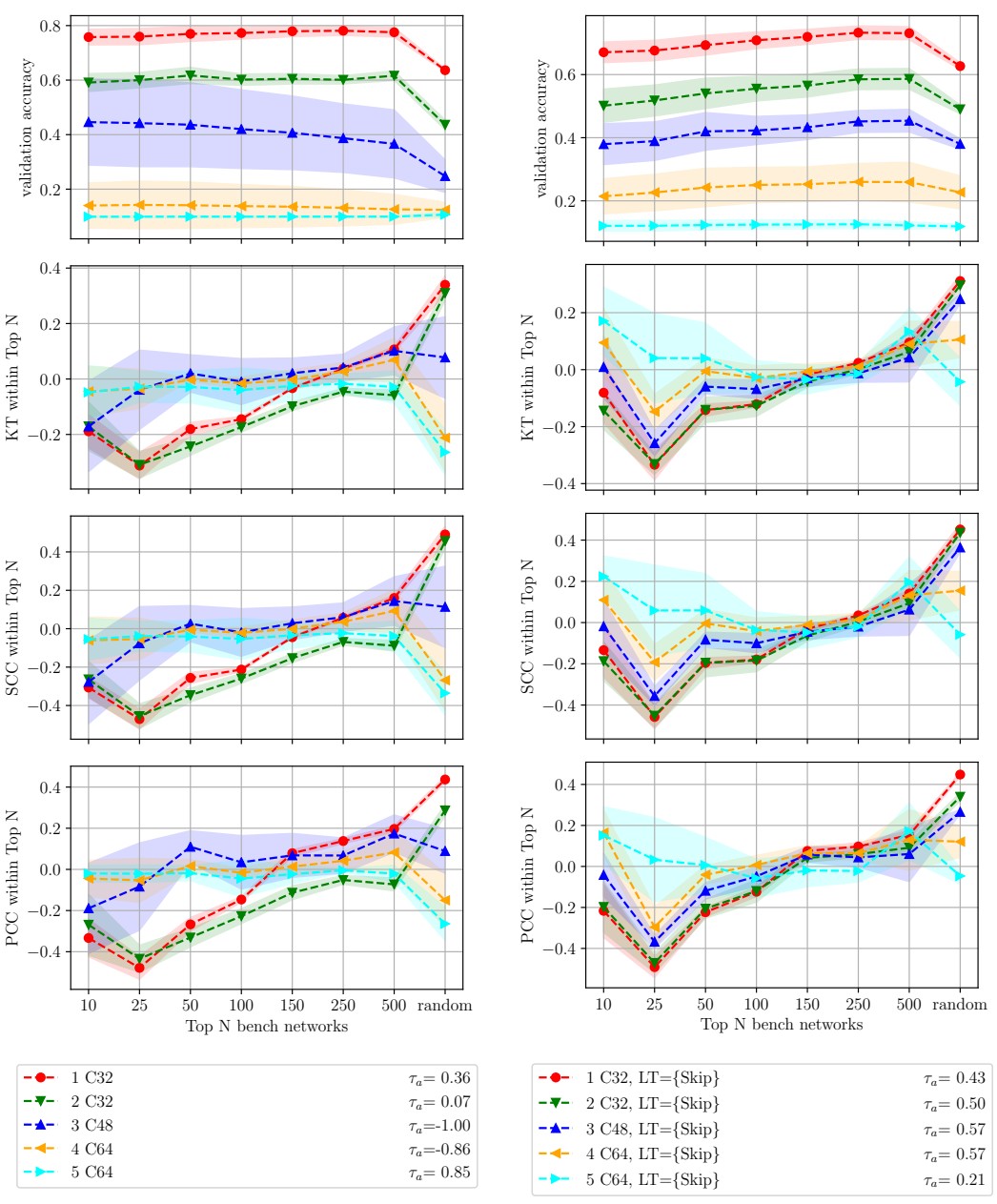

Figure 14: Super-networks with 1 to 5 normal cells per stage. Smaller sized weight-sharing super-networks are generally easier to train and better predictors in the full search space.

