# OpenReview forum: "Exploring single-path Architecture Search ranking correlations"
_ICLR.cc/2021/Conference — Reject_

### Official Review · AnonReviewer1 · 2020-10-13

**Rating:** 5
**Confidence:** 4

**Review:**

In this paper, the authors proposed several findings on the single-path training strategy. The ranking correlation is the main issue. The experiments are conducted on NASBench201.

Introduction.

'However, the aforementioned weight sharing xxx', there are a number of efficient multi-objective (Pareto front) NAS methods.

'However, since the single-path', please cite some literature related to the ranking issue.

Summarize the main findings in the introduction section.

Method.

Define $\tau_{\alpha}$ in math. 'describing the ranking correlation of the average prediction accuracy depending on N', what is the average prediction accuracy for two lists?

Experiments.

4.1 'masking the Zero operation (bottom row) significantly reduces this portion and thus improves the ranking correlation $\tau$ (KT)'. If removing other operations other than zero, will the $\tau$ be lower? In my opinion, removing any operation leads to smaller search space, and they all have a higher ranking.

4.2, 4.2 examine some of the training strategies proposed by previous works.

References.

Please make the references clear. Add the venues for all the papers.




This paper tries to explore the single-path training strategy by studying the search space, the supernet, the linear transformer, the strict uniform sampling, the topology sharing, the LR warmup, the regularization, the clipping. The authors have done lots of experiments to clarify the important reasons for the ranking. However, most of the findings are not new to me. They have been discussed more or less by previous works and discovered by my own experiments. So the contribution is not significant. The paper is mostly clear, some paragraphs and references need to be polished, more related works should be added.

---

> ### Author Response · Authors · 2020-11-13
> **Rebuttal reply R1**
>
>
> We thank you for the invested time in reviewing our work,
> and the suggestions that help us making the paper more accessible.
>
>
>
> - 'However, the aforementioned weight sharing xxx', there are a number of efficient multi-objective (Pareto front) NAS methods.
>     - That is absolutely correct, and the SPOS-based [A1] method that we use can be used for multi-objective optimization
>       with hardly any additional effort. However, if optimizing a single target is already hard
>       (accuracy is arguably the most important one), as we can see in our
>       experiments, adding additional targets will not give us any benefits.
>     - Edit to clarify (Nov. 15): "no benefits" with respect to the first target. Naturally, if the second target is e.g. latency,
>       a multi-objective search will choose fast architectures. But as the targets do not depend on each other, we can study
>       them in isolation. Furthermore, any improvement on the accuracy prediction also benefits multi-objective
>       methods.
>
>
> - 'However, since the single-path', please cite some literature related to the ranking issue.
>     - We added the SPOS [A1], FairNAS [A2], and SCARLET-NAS [A3], which we believe to be good examples.
>
>
> - Summarize the main findings in the introduction section.
>     - We added a paragraph at the end of the introduction section that briefly outlines the paper structure and includes
>     our most important findings. We feel that this is a strong improvement in accessibility and are grateful
>     for the suggestion.
>
>
> - Define \tau_a in math. 'describing the ranking correlation of the average prediction accuracy depending on N',
> what is the average prediction accuracy for two lists?
>     - We have added a formal description that hopefully answers your question
>     (due to formatting we do not cite it here).
>
>
> - 4.1 'masking the Zero operation (bottom row) significantly reduces this portion and thus improves the ranking correlation
> (KT)'. If removing other operations other than zero, will the \tau be lower?
> In my opinion, removing any operation leads to smaller search space, and they all have a higher ranking.
>     - Given the Bench-201 set of operations, there are some rare exceptions. Masking e.g. the Zero operation
>     (see Figure 4, top left plot, top-10 networks) is actually detrimental.
>     However, observing \tau in the same Figure seconds your point, a smaller search space is easier to rank
>     (the baseline \tau is often worse than those of space subsets).
>     - We further want to point out that the search space subsets are a means to evaluate the robustness of search variations,
>     such as adding linear transformers [A3]. Being able to also validate the few-shot approach [A4] with these experiments
>     is a bonus to us, but not the main intention.
>
>
> - 4.2 examine some of the training strategies proposed by previous works.
>     - The experiments cover SPOS [A1], FairNAS sampling [A2] (default in the experiments), linear transformers [A3],
>     multiple regularization techniques, the novel approach of disabling cell-topology-sharing during the search, and
>     indirectly also few-shot search [A4].
>     Please point us to any interesting or important topic that you feel we left uncovered.
>     - Please clarify if we understood you wrong, and you meant us to e.g. further elaborate on the super-net training process,
>     so that we can make the appropriate changes in time.
>
>
> - Please make the references clear. Add the venues for all the papers.
>     - You are right, it is only good courtesy to add them as they are available. We have added all the venues we could
>     find via Google Scholar.
>
>
> Sincerely,
>
> First author
>
>
> [A1] Single Path One-Shot Neural Architecture Search with Uniform Sampling
>
> [A2] FairNAS: Rethinking Evaluation Fairness of Weight Sharing Neural Architecture Search
>
> [A3] SCARLET-NAS: Bridging the gap between Stability and Scalability in Weight-sharing Neural Architecture Search
>
> [A4] Few-shot Neural Architecture Search

---

> ### Author Response · Authors · 2020-11-15
> **Rebuttal reply R1 part 2**
>
> (since we hit the character limit, this is part 2 of our reply)
>
>
> -  However, most of the findings are not new to me. They have been discussed more or less by previous works and discovered by my own experiments. So the contribution is not significant.
>     - We are happy to read that our results support your own findings, but it seems we failed to communicate
>     the novel parts in our work properly:
>          - we experiment with the idea of disabling the topology sharing only during training time
>          - the experiment design itself is novel, as we measure the ranking correlations in
>          many top-N network groups, which is only made possible by having extensive knowledge of the search space.
>          The thus obtained results give us insight not only in how well randomly sampled architectures are ranked,
>          but also if they still work in the subset of the top-500 networks (top <3% of the search space).
>     - And present interesting results, which we have not seen in literature yet:
>          - we study the effectiveness of method variations using the same environment (training schedule, network, data ...)
>          and in multiple search spaces. As many presented methods vary in this respect, or are limited to only one search
>          space, we saw a need in a broader and fair study.
>          - seconding the above point, we find that linear transformers [A1] only work in their proposed environment
>          and are otherwise harmful
>          - we also find that common training variations (regularization, learning rate warmup, gradient clipping)
>          do not improve the ranking correlations, independent of the search space
>     - We hope that the newly added paragraph in the introduction Section states our contribution more clearly, and that we could convince you, both of the necessity and the gained insights of a fair and broad study.
>
>
>  - The paper is mostly clear, some paragraphs and references need to be polished, more related works should be added.
>     - We have added additional related work, references, some text (see our "Changes to the paper" post), and added
>     venues to the references in our newly uploaded version.
>     Please let us know if we neglected to mention further specific references or details.
>
>
>  Sincerely
>
>  First author
>
>  [A1] SCARLET-NAS: Bridging the gap between Stability and Scalability in Weight-sharing Neural Architecture Search

---

### Official Review · AnonReviewer3 · 2020-10-27
**A timely analysis of the current NAS's ineffectiveness caused by the inaccurate architecture rating problem**

**Rating:** 8
**Confidence:** 5

**Review:**


+ This paper studies the single-path one-shot super-network predictions and ranking correlation throughout an entire search space, as all stand-alone model results are known in advance. This is a crucial step in NAS. As we know, inaccurate architecture rating is the cause of ineffective NAS in almost all existing NAS methods. It makes nearly all previous NAS methods not better the random architecture selection (suggested by two ICLR 2020 papers and many ICLR 2021 submissions). Therefore, analyzing the architecture rating problem is of most importance in NAS. This paper takes a deep insight into the architecture rating problem, which provides a timely metric for evaluating NAS's effectiveness. (+)


- In the following text, another paper entitled "Block-wisely Supervised Neural Architecture Search with Knowledge Distillation" should be discussed: "Recent efforts have shown improvements by strictly fair operation sampling in the super-network training phase (Chu et al. (2019b)), by adding a linear 1×1 convolution to skip connections, improving training stability (Chu et al. (2019a)), or by dividing the search space (Zhao et al. (2020))," (-)


- Kendall's Tau is a good metric. As shown in EagleEye, Spearman Correlation Coefficient(SCC) and Pearson Correlation Coefficient (PCC) are also good metrics. Could the authors also provide a comparison using these two metrics? (-)

EagleEye: Fast Sub-net Evaluation for Efficient Neural Network Pruning


- I think NAS-Bench-201 is not enough. As we know, CIFAR-10 is sometimes considered a toy benchmark, and the sole result on CIFAR-10 is not convincing. Could the authors provide more results in addition to CIFAR-10? (-)


- As we know, there may be a gap between the small-channel supernet and the large-channel finally-adopted architecture. We are quite interested in the ranking correlations between a subnet obtained from the small-channel supernet and a channel-expanded version of the subnet trained from scratch. Could the authors provide such a ranking correlation analysis? (-)


- Could the authors provide more details in Figure 3. Figure 3 shows that the lines on the top mean the operation is used more frequently. But I am not sure what the value of the y-axis means. (-)


- Could the authors present some comments on "Perhaps the most surprising is the low importance of Average Pooling, even lower than Zero, an operation that does absolutely nothing"? (-)


+ The following observation is believed to be crucial in NAS: "The baseline for small networks (top left, red) has the same averaged prediction accuracy for the top 10 as for the top 500 networks". This validates the inefficiency of SPOS in architecture search. (+)


+ The following observation is also important in NAS: "Masking Skip (blue, left) is the most harmful to τa (=1). As seen in Figure 4, the top-N networks have a worse average predicted accuracy than the top-M (for N < M) networks, and sometimes even below the random sample, which is terrible. Interestingly, \tau may improve within the predictions for the top-N architectures." Especially, the phenomenon that masking skip connection reduces the ranking correlations is interesting. As is shown in SCARLET-NAS, the supernet training with skip connection is not fair. But in this paper, we can see that skip connection benefits the ranking correlation. We are interested in this opposite opinion. Specifically, it is fascinating to see that "Although the additional transformers seem to stabilize training, as seen by the lower standard deviation, they also worsen the τa problem." Besides, the phenomenon of "\tao may improve within the predictions for the top-N architectures" indicates that the metric for ranking correlations maybe not perfect. A more reasonable metric may be desirable. (+-)


+ The following observation is important: "medium-sized super-networks require additional care." As shown by Figure 4, the averaged predicted accuracy of top-N networks in several subsets is lower than that of a random subset of networks. This is consistent with previous work like DNA, which shows a large search space may be harmful to the architecture rating. Even if a medium-sized supernet has a bad architecture rating, the ranking correlation should be worse in a large-sized supernet. (+)

DNA: Block-wisely Supervised Neural Architecture Search with Knowledge Distillation


- The following description is questionable: "After the architecture search, all Linear Transformers can safely be removed, as they do not impact the network capacity". Actually, stacking many fully connected layers without non-linear activations could lead to only one fully connected layer. It is an open question of whether optimizing loss(ABCx, y) is as difficult as optimizing loss(Dx, y) using stochastic gradient descent. (-)


+ The results providing evidence against disabling cell topology sharing during the training phase are exciting and new to the public. (+)


+ The following observation is fascinating: "The absolute validation accuracy value is increased by uniform sampling. However, this is not relevant, as only the correct ranking matters". This is against FairNAS. (+)


+ It is interesting and convincing that many tricks such as learning rate warm-up, gradient clipping, and regularization do not work to improve the ranking correlation. We are pleased that the authors provide so many experiments to point out some misleading approaches in NAS. I think this paper is very important in the context of AutoML. (+)


- The analysis is based on medium-sized and small-sized search space. It would be good to see some analysis of large-sized search space. (-)


Overall, this paper provides a timely analysis of the current NAS's ineffectiveness caused by the inaccurate architecture rating problem. As there are many NAS papers published every year and their ineffectiveness may still be not widely recognized by the reviewers and the public, I recommend a strong acceptance for this paper to promote the analysis of the NAS's architecture rating problem.




------------------------------post rebuttal------------------------------------------


-------------------Response to the authors' response----------------------


Thank you for the hard work in responding.

I have read other reviewers' reviews and the response from the authors. The authors have addressed most of my concerns.

I believe this paper deserves acceptance. As we know, variants of efforts have been made to improve NAS's effectiveness since 2016, and a great process has been reached. Despite the high expectation and solemn devotion, NAS's effectiveness is believed to be still low. This is inconsistent with many pioneer researchers' expectations four years ago, in which NAS is expected to be another revolutionary technique similar to 2012's deep learning. Currently, there are many NAS papers published every year. But their effectivenesses are unclear due to the lack of ranking correlation analysis. Differently, this paper comprehensively analyzes the architecture rating problem, which provides a timely analysis of the current NAS's ineffectiveness caused by inaccurate architecture rating. I think this paper can attract the community's attention, encouraging the community to pay attention to the architecture rating in NAS, especially when reviewing a NAS paper. Therefore, I recommend an acceptance for this paper to promote the analysis of the NAS's architecture rating problem.

I agree with R2 that Yu et al. have proposed a similar idea (I assume R2 refers to "Kaicheng Yu, Christian Sciuto, Martin Jaggi, Claudiu Musat, Mathieu Salzmann, Evaluating the Search Phase of Neural Architecture Search"). But the analysis in this paper is more comprehensive than Yu et al.'s article. Many findings are new (at least they are not in published papers).

I agree with R4 that the authors did not form a coherent logic flow to present these empirical findings, and the paper was similar to a technique report. However, many important articles, e.g., "Designing Network Design Spaces," "Exploring Simple Siamese Representation Learning," "Is Faster R-CNN Doing Well for Pedestrian Detection?" are also technique-report-like.

I appreciate R1 for his devotion to finding similar observations in his experiments. I believe these observations are important and deserve publication. I agree with R1 that removing any operation leads to a smaller search space and a higher ranking.

In summary, I will keep my rating as an acceptance.

Undoubtedly, I also believe the comments from other reviewers can benefit the improvement of your paper.

---

> ### Author Response · Authors · 2020-11-13
> **Rebuttal reply R3 part 1**
>
> (due to the character limit, we have to split our reply into parts)
>
> Thank you for your kind words of encouragement and the many great points that you make.
>
> It is a hard task to keep track of the currently best NAS methods, given that they use different search spaces,
> data sets, regularization methods, optimizers, learning rate schedules, batch sizes, training epochs,
> specifics such as limiting the number of skip connections, and so on.
> We therefore greatly appreciate benchmarks such as Bench-201, which, combined with the rather cheap evaluation of
> SPOS super-network predictions, enable a systematic and thorough comparison.
>
>
> - In the following text, another paper entitled
> "Block-wisely Supervised Neural Architecture Search with Knowledge Distillation"
> should be discussed: "Recent efforts have shown improvements by strictly fair
> operation sampling in the super-network training phase (Chu et al. (2019b)),
> by adding a linear 1×1 convolution to skip connections, improving training stability (Chu et al. (2019a)),
> or by dividing the search space (Zhao et al. (2020))," (-)
>     - You are correct, in a sense DNA also divides the search space, albeit in another way than few-shot learning.
>     We have not considered using their technique, since that introduces the question how shared cell topologies should
>     be handled, which may be a topic on its own. However we added the reference to the text, which has been changed:
>         - *"Recent efforts have
>         shown improvements by strictly fair operation sampling in the super-network training phase (Chu
>         et al. (2019b)) and adding a linear 1×1 convolution to skip connections, improving training stability
>         (Chu et al. (2019a)). Other works divide the search space, exploring multiple models with different
>         operation-subsets (Zhao et al. (2020)), or one model with several smaller blocks that use a trained
>         teacher as a guiding signal (Li et al. (2020))."*
>
>
> - Kendall's Tau is a good metric. As shown in EagleEye, Spearman Correlation Coefficient(SCC)
> and Pearson Correlation Coefficient (PCC) are also good metrics.
> Could the authors also provide a comparison using these two metrics? (-)
>     - Absolutely. We have added Appendix D, where we selected some of the experiments and re-evaluated them,
>     adding SCC and PCC.
>
>
> - I think NAS-Bench-201 is not enough. As we know, CIFAR-10 is sometimes considered a toy benchmark,
> and the sole result on CIFAR-10 is not convincing. Could the authors provide more results in addition
> to CIFAR-10? (-)
>     - The computational cost of super-net training and especially acquiring ground-truth data on e.g. ImageNet is prohibitively
>     expensive, so that experiment typically only compare 10-20 models to measure \tau (see e.g. FairNAS [A1]).
>     Such a comparison lacks information about the entire search space, such as knowing how good the discovered models
>     *actually* are - their ranking may be correct (high \tau), but the models may still not be within the top 10%.
>     However, the recent idea of surrogate benchmarks [A2] may make such a study possible in the near future.
>
>
> - As we know, there may be a gap between the small-channel supernet and the large-channel
> finally-adopted architecture. We are quite interested in the ranking correlations between a
> subnet obtained from the small-channel supernet and a channel-expanded version of
> the subnet trained from scratch. Could the authors provide such a ranking correlation analysis? (-)
>     - While we find this idea very tempting, we will not be able to train the amount of networks required for a good
>     analysis in time. Instead, we have trained small networks starting with 96 channels (as opposed to 32),
>     which is much closer to the final width.
>     These experiments also include the Pearson and Spearman metrics,
>     and are also listed in Appendix D.
>
>
> - Could the authors provide more details in Figure 3. Figure 3 shows that the lines on
> the top mean the operation is used more frequently.
> But I am not sure what the value of the y-axis means. (-)
>     - We have added the following text to the description of Figure 3., which hopefully answers the question:
>         - *"As an example, if 3 of the top-10 networks use at least one Pool operation,
>     together a total of 4 Pool operations, and have all 5 operations available,
>     the usage is $3/10 = 0.3$ while the share is $4/(5\cdot10) = 0.08$."*

---

> > ### Author Response · Authors · 2020-11-17
> > **Rebuttal reply R3 part 1, SSC/PCC**
> >
> > We have updated the paper with the additional metrics SCC and PCC, please see the new Appendix D.
> > The experiments were selected from some prior ones, and additional small but wide super-networks (2 normal cells per stage, starting with 96 instead of 32 channels).
> >
> > Our github repository has been updated to include SCC and PCC.
> >
> >
> >
> > Sincerely,
> >
> > First Author

---

> > > ### Author Response · Authors · 2020-11-19
> > > **Rebuttal reply R3 part 1, number of normal cells**
> > >
> > > We have added further plots for super-networks with N=1 to 5 normal cells per stage to Appendix D, and find that in the full search space, a smaller network is a better predictor.
> > >
> > > Sincerely,
> > >
> > > First Author

---

> ### Author Response · Authors · 2020-11-13
> **Rebuttal reply R3 part 2**
>
>
> - Could the authors present some comments on "Perhaps the most surprising is
> the low importance of Average Pooling, even lower than Zero,
> an operation that does absolutely nothing"? (-)
>     - We have altered the text to the following:
>         - *"Perhaps the most surprising is the low importance of Average Pooling, even lower than Zero.
>     It appears that all the benefits of Pool are already covered by the $3\times3$ Convolution,
>     so that using the unnecessary operation now decreases the network accuracy."*
>
>
> - The following observation is believed to be crucial in NAS:
> "The baseline for small networks (top left, red) has the same averaged
> prediction accuracy for the top 10 as for the top 500 networks".
> This validates the inefficiency of SPOS in architecture search. (+)
>     - Yes and no. We can also see that the top-500 models have a much better average super-net accuracy than a
>     random pick over the entire search space, and \tau is between 0.2 and 0.4 in most sub-spaces.
>     It is not trivial to answer whether that is also accurate in e.g. the SPOS or FairNAS ImageNet search spaces,
>     which lack harmful operations such as Zero or Pool, which are also rarely used in the given top-500 networks -
>     this remains an interesting question for future research.
>
>
> - Especially, the phenomenon that masking skip connection reduces the ranking correlations is interesting.
> s is shown in SCARLET-NAS, the supernet training with skip connection is not fair.
> But in this paper, we can see that skip connection benefits the ranking correlation.
> We are interested in this opposite opinion.
>     - We believe this to be the result of the different search space. By removing the skip connection, many top-N
>     Bench-201 networks make use of Average Pooling (Figure 3., top right, top plot), likely since the fully sized
>     standalone model training benefits from the non-parametrized paths.
>     In contrast, the super-networks have troubles with ranking these models correctly.
>
>
> - Specifically, it is fascinating to see that
> "Although the additional transformers seem to stabilize training,
> as seen by the lower standard deviation, they also worsen the τa problem."
> Besides, the phenomenon of "\tao may improve within the predictions for the top-N architectures"
> indicates that the metric for ranking correlations maybe not perfect.
> A more reasonable metric may be desirable. (+-)
>     - We also consider the behavior of linear transformers very interesting, especially that it's not advisable to
>     simply add them regardless of the search space
>     (and the same is true for regularization methods, as pointed out further down).
>     Granted, they do work decently within their proposed domain [A3],
>     it is simply that experiments in other search spaces were not considered.
>
>
> - The following observation is important: "medium-sized super-networks require additional care."
> As shown by Figure 4, the averaged predicted accuracy of top-N networks in several
> subsets is lower than that of a random subset of networks.
> This is consistent with previous work like DNA,
> which shows a large search space may be harmful to the architecture rating.
> Even if a medium-sized supernet has a bad architecture rating,
> the ranking correlation should be worse in a large-sized supernet. (+)
>     - Interestingly, this conflicts with P-DARTS [A4], in which the authors find that a larger super-network bridges
>     the evaluation gap between super-network and standalone model.
>     However, that is quite likely due to the difference of single-path and gradient-based training, as a gradient-trained
>     NAS algorithm already converges into the final architecture during training, which single-path methods do not
>     (except for operation pruning).
>
>
> - The following description is questionable: "After the architecture search,
> all Linear Transformers can safely be removed, as they do not impact the network capacity".
> Actually, stacking many fully connected layers without non-linear activations
> could lead to only one fully connected layer.
> It is an open question of whether optimizing loss(ABCx, y)
> is as difficult as optimizing loss(Dx, y) using stochastic gradient descent. (-)
>     - Good catch. While the capacity remains the same, your point about the network training is valid.
>     We have rephrased the sentence the avoid a misleading interpretation:
>         - *"All Linear Transformers are removed after the search, resulting in a standalone
>         network with the same capacity."*

---

> ### Author Response · Authors · 2020-11-13
> **Rebuttal reply R3 part 3**
>
>
> - It is interesting and convincing that many tricks such as learning rate warm-up,
> gradient clipping, and regularization do not work to improve the ranking correlation.
> We are pleased that the authors provide so many experiments to point out some misleading approaches in NAS.
> I think this paper is very important in the context of AutoML. (+)
>     - This also surprised us, given that regularization is very common addition to super-network training.
>     In search spaces where all operations can benefit from a given technique, they are probably not harmful though
>     (but not necessarily improving the ranking).
>
>
> - The analysis is based on medium-sized and small-sized search space.
> It would be good to see some analysis of large-sized search space. (-)
>     - Unfortunately, models with 6 normal cells per stage and 64 channels require too much GPU memory under the same
>     training schedule (i.e. batch size and channel count) on our Nvidia 1080 Ti GPUs.
>     However, we can evaluate models with 1 to 5 normal cells per stage if you are interested in such a comparison.
>
>
> Sincerely,
>
> First author
>
>
> [A1] FairNAS: Rethinking Evaluation Fairness of Weight Sharing Neural Architecture Search
>
> [A2] NAS-Bench-301 and the Case for Surrogate Benchmarks for Neural Architecture Search
>
> [A3] SCARLET-NAS: Bridging the gap between Stability and Scalability in Weight-sharing Neural Architecture Search
>
> [A4] Progressive Differentiable Architecture Search: Bridging the Depth Gap between Search and Evaluation

---

### Official Review · AnonReviewer4 · 2020-10-28
**An empirical study on the ranking correlation of several NAS mechanisms in the singe-path setup**

**Rating:** 5
**Confidence:** 4

**Review:**

This paper introduces an empirical study on the ranking correlation in the singe-path setup. Following the paradigm of NAS-Bench-201,  the authors test the Kendall Tau correlation of networks from NAS training and networks from standalone training.

In general, I appreciate the authors' effort in bringing more infrastructure to the community of NAS. As a recently emerged community, we do need works like this one, as well as previous ones such as NAS-Bench-101 and NAS-Bench-201, to make the evaluation protocol more scientific. NAS problems are non-trivial as the search space is notoriously large. Colleagues who would like to invest their time and resources in exploring and manifesting this search space to uncover more phenomena are thus worth respect.

However, this respectable responsibility also comes with a higher standard to evaluate works attempting to fulfill it. My major concern with this work is that the manuscript is not organized well. Although authors provide substantial details on their empirical study, they did not form a coherent logic flow to present these empirical findings, which makes this work more like a technical report than an academic paper. Readers may find these phenomena interesting but may not get interesting insights after reading this paper. Hence the technical contribution, especially on novelty, seems quite limited, even if there may be some intriguing points in the authors' discovery. I would recommend the authors to pick some phenomena e.g., masking Zero, masking Skip, etc., as examples to provide more analysis, so as to demonstrate to colleagues in our community that these findings can indeed lead to interesting research topics.

Some minors: There are some works missed in the literature review. For example, the authors did not give adequate credits to colleagues who pioneered in using Kendall Tau to evaluate NAS training. As far as I know, Sciuto et al., 2019 was one of the earliest works. In the third paragraph, when reviewing recent progress, the authors did not distinguish the ranking correlation between NAS searching and retraining from the correlation between NAS searching results and stand-alone training. The former one was discussed and addressed in Hu et al., 2020. As the community has not fully realized the subtle but crucial difference between these two correlations, I believe a better framing of this work can be more helpful to other colleagues, especially those new comers.

Sciuto et al. 2019, Evaluating the search phase of neural architecture search.

Hu et al. 2020, DSNAS: Direct Neural Architecture Search without Parameter Retraining

---

> ### Author Response · Authors · 2020-11-13
> **Rebuttal reply R4**
>
> Thank you for investing your time reviewing our work
> and pointing out many of our paper's shortcomings, which is a great help in improving it.
>
>
> - However, this respectable responsibility also comes with a higher standard to evaluate
> works attempting to fulfill it. My major concern with this work is that the manuscript
> is not organized well. Although authors provide substantial details on their empirical study,
> they did not form a coherent logic flow to present these empirical findings, which makes
> this work more like a technical report than an academic paper.
>     - The lack of logic flow is a valid concern,
>     given that we explore multiple recent variations to SPOS [A1] and some regularization techniques,
>     rather than focusing on a single detail or method.
>     As we have added a small overview that includes an outlook to the results on behalf of another reviewer, we hope that
>     this particular concern has also been addressed.
>     If not, please provide us with feedback on how to improve the readability of this work.
>
>
> - Readers may find these phenomena interesting but may not get interesting insights after
> reading this paper. Hence the technical contribution, especially on novelty, seems quite limited,
> even if there may be some intriguing points in the authors' discovery.
>     - We explore many already-presented methods, which are naturally not novel by any means. However, we also
>         - use SPOS-based methods [A1] to study the ranking correlations in many top-N network groups, which is a novel
>         approach to evaluate the effectiveness of method variations
>         - experiment with disabling the topology sharing only during training time,
>         an attempt that we have not yet seen in literature
>         - find that linear transformers [A2] only work in their proposed environment and are otherwise harmful
>         - find that many commonly used training variations (regularization, learning rate warmup, gradient clipping)
>         actually have little to no benefit for the ranking correlation, which is a very interesting insight
>     - Nonetheless, it appears we failed to properly communicate the points above in our paper.
>     As we have now briefly included them in the introduction section as mentioned in the point above, we hope that
>     the paper is now easier to follow in structure more interesting to read.
>     Please do not hesitate to voice your concerns if you feel that we have not properly addressed your point.
>
>
> - I would recommend the authors to pick some phenomena e.g., masking Zero, masking Skip, etc.,
> as examples to provide more analysis, so as to demonstrate to colleagues in our community
> that these findings can indeed lead to interesting research topics.
>     - Our main intent was to evaluate training variations that have been proposed across different search spaces
>     and under different conditions, but not actually been compared in effectiveness.
>     We fully agree that new research topics are often more exciting than an experimental study of known techniques,
>     nevertheless we hope that our findings have sparked further interest comparing some of the many recently proposed
>     approaches in a thorough and fair way.
>
>
> - For example, the authors did not give adequate credits to colleagues
> who pioneered in using Kendall Tau to evaluate NAS training.
> As far as I know, Sciuto et al., 2019 was one of the earliest works.
>     - You are correct that we should add credit where it is due and
>     add Sciuto et al. and FairNAS [A3] as references, as we are not aware of any other work that predates
>     Sciuto et al. using Kendall Tau for NAS ourselves.
>
>
> - In the third paragraph, when reviewing recent progress,
> the authors did not distinguish the ranking correlation between NAS searching and retraining
> from the correlation between NAS searching results and stand-alone training.
> The former one was discussed and addressed in Hu et al., 2020.
> As the community has not fully realized the subtle but crucial difference between
> these two correlations, I believe a better framing of this work can be more helpful
> to other colleagues, especially those new comers.
>     - We have added the following sentence that clarifies that reusing search network weights is not a given:
>         - *"Although the standalone network is often trained from scratch, reusing the search network weights can
>         increase both training speed and final accuracy (citing [A4], DSNAS)."*
>
>
> Sincerely,
>
> First author
>
>
> [A1] Single Path One-Shot Neural Architecture Search with Uniform Sampling
>
> [A2] SCARLET-NAS: Bridging the gap between Stability and Scalability in Weight-sharing Neural Architecture Search
>
> [A3] FairNAS: Rethinking Evaluation Fairness of Weight Sharing Neural Architecture Search
>
> [A4] HM-NAS: Efficient Neural Architecture Search via Hierarchical Masking

---

> > ### Comment · AnonReviewer4 · 2020-11-24
> > **Some examplar works for empirical study**
> >
> > Thank you for your response.
> >
> > During the discussion among reviewers, R3 raised some very good previous works in formulating empirical study, which I believe can definitely inspire you in your reiteration:
> >
> > >"Designing Network Design Spaces," "Exploring Simple Siamese Representation Learning," "Is Faster R-CNN Doing Well for Pedestrian Detection?"
> >
> > I believe this work can hopefully be accepted in a future venue if you improve its organization towards these exemplars. For the current version, I would stand my initial rating.

---

### Official Review · AnonReviewer2 · 2020-10-29
**Official Blind Review #2**

**Rating:** 5
**Confidence:** 4

**Review:**

This paper studies the relationship of correlation of ranking of networks sampled from SuperNet and that of stand-alone networks under various settings. They also study the how masking some operations in the search space and different ways of training effect the ranking correlation.

Pros:
The paper has a lot of experiments to substantiate the claims.
Figure 3 where every operation is systematically masked, provides more insights about which operations are effective and how NAS behaves if one of the operation is masked.

Cons:
Several other papers have already published similar findings. Overall the paper is very incremental.
More specifics in the questions

Questions

1. How is the SuperNet trained?
2. Figure2: Yu et al [1] have already explored the correlation of ranks of networks sampled from SuperNet and that of stand-alone networks. How is Figure 2 different from that?
3. RobustDarts [2] has explored the possibility of how subset of NASBENCH search spaces behave. FAIRDarts [3] also explored the influence of skip connection by running DARTS without skip connection, running random search by limiting skip connection to 2 etc. Figure 4 seems to be inspired by that. While it is interesting, this might be a slight extension to the work done by Yu et al [1]
4. Bender et al [4] postulate that the operations of a SuperNet are subject to co-adaptation and recommended techniques such as regularization, drop path etc to alleviate the same. RobustDarts also suggest some recommendations such as L2 regularization, drop path etc although in the context of DARTS. So while Figure 6 demonstrates this empirically, it is not a new finding.

Overall, the empirical results in the paper are very useful for the NAS community. But the work is still very incremental. This might be better received as a workshop paper instead.

---

> ### Author Response · Authors · 2020-11-13
> **Rebuttal reply R2**
>
> We thank you for the time invested in providing the comment,
> and in pointing out many interesting papers that investigate similar properties in gradient-based algorithms.
> We hope that our answers to your concerns can clarify the issues at hand. If not, please feel free to ask.
>
> Please add a link for reference [1], as we can not see it and are uncertain which paper it refers to.
> If you feel that we have missed the point of your related questions, we will then clarify the issues.
>
>
> - How is the SuperNet trained?
>     - You can find a description in Section 3.3 and in Appendix A.
>     We have also recently released the code base used for all experiments, which hopefully answers your question.
>     If not, do not hesitate to ask us about further specifics.
>
>
> - Figure2: Yu et al [1] have already explored the correlation
> of ranks of networks sampled from SuperNet and that of stand-alone networks.
> How is Figure 2 different from that?
>     - You are correct that Figure 2. is not novel by any means,
>     but only intended to give a visual intuition about the experiment method
>     and what Figures 4. to 6. condense to single data points.
>
>
> - RobustDarts [2] has explored the possibility of how subset of NASBENCH
> search spaces behave. FAIRDarts [3] also explored the influence of skip connection
> by running DARTS without skip connection, running random search by limiting skip
> connection to 2 etc. Figure 4 seems to be inspired by that. While it is interesting,
> this might be a slight extension to the work done by Yu et al [1]
>     - Both RobustDARTS and FairDARTS are using a gradient-based architecture search method, which has different properties
>     than the SPOS-based [A1] strategies that we experiment with.
>         - While both works are very interesting in the former domain, the design of SPOS permits us to evaluate many more networks
>         than we actually have to train, which is very different from gradient-based methods that converge to exactly one solution.
>         - In our current understanding, a gradient-trained network can not be used in the way that we experiment with:
>         estimating the super-network accuracy for any particular architecture.
>         Instead, gradient-based ranking comparisons are often limited to a small number of trained super-networks and their
>         respective stand-alone results, where hopefully the ranking of super-network accuracies and stand-alone accuracies match.
>         This does still not examine the ranking correlation over an entire search space (or top-N subsets thereof),
>         even with ground-truth stand-alone results we can only estimate how the results of these methods are distributed
>         (i.e. the probability to converge to a top-1% model).
>
>
> - Bender et al [4] postulate that the operations of a SuperNet are subject
> to co-adaptation and recommended techniques such as regularization,
> drop path etc to alleviate the same. RobustDarts also suggest some
> recommendations such as L2 regularization, drop path etc although in
> the context of DARTS. So while Figure 6 demonstrates this empirically, it is not a new finding.
>     - Figure 6. actually highlights the ineffectiveness of many regularization techniques.
>     We can see that, almost independent of the search space and technique, the top-N network predictions in small search networks
>     (for N=10, 25, ...500) behave nearly the same.
>     We have not experimented with path dropout, given that the search space already has a Zero operation and this
>     technique is probably much less useful in more linear cell-designs, where it is equivalent to block-drop [A2].
>     - [A3] experimented with adding drop-path to only Skip operations.
>     Systematically exploring the effectiveness of such an approach,
>     given different operations of which only certain ones can be dropped,
>     may provide interesting insights in future works.
>
>
> Sincerely,
>
> First author
>
>
> [A1] Single Path One-Shot Neural Architecture Search with Uniform Sampling
>
> [A2] BlockDrop: Dynamic Inference Paths in Residual Networks
>
> [A3] Progressive Differentiable Architecture Search: Bridging the Depth Gap between Search and Evaluation

---

### Author Response · Authors · 2020-11-09
**Code to reproduce**

Dear reviewers and readers,


thank You for the time already spent with our work. Since reproducing results is often an issue, we decided to make the the code available after some cleaning-up.

The link destination is not anonymized.
There are some instructions on the page to how to run the experiments.
<removed>


Sincerely,
First author

---

> ### Comment · Program_Chairs · 2020-11-17
> **non-anonymized  links are not permited**
>
> Dear authors,
>
> Links to unanonymized resources  are not permitted.
>
> - program chairs.

---

> > ### Author Response · Authors · 2020-11-17
> > **Code to reproduce**
> >
> > We apologize for incautiousness and have uploaded the zipped code in the supplementary material instead.
> >
> > Sincerely,
> >
> > First author

---

### Author Response · Authors · 2020-11-13
**Changes to the paper**

Dear reviewers,

thank you for evaluating our work and helping us in improving it. According to your feedback, we have made the following changes to the paper:

---
(Date: November 13, 2020)


Abstract
- minor rephrasing

Introduction:
- added text about retraining with search-net weights
- added citations for single-path methods
- added overview/summary paragraph at the end

Related work
- added DNA

Figure 3
- added clarification text

Section 3.4
- added formal description of \tau_a

Section 4.1
- added text about Zero/Pool importance

Section 4.2
- removed that linear transformers can be "safely" removed, even though the capacity is the same

Appendix D
- added additional plots including the SCC and PCC metrics

References
- added venues


---
(Date: November 17, 2020)

Appendix D
- updated the plots


---
(Date: November 19, 2020)


Section 3.1
- mention Appendix D

Conclusions
- mention Appendix D

Appendix D
- added plots for networks using a different number of normal cells


---

Further changes to the paper will extend the above list, with a respective timestamp.

We have not yet updated our github repository (see the anonymized link in another comment), but plan to do so in the coming days.

Sincerely,

First author

---

### Decision · Program_Chairs · 2021-01-07
**Final Decision**

**Decision:**

Reject

**Comment:**

This submission received reviews with a very wide range of scores (initially 3,5,5,9; then 5,5,5,9). In the discussion, all reviewers maintained their general position (although a private message by the reviewer giving a score of 9 said he/she would consider going down to an 8).

Because of the high variance, I read the paper in detail myself. I agree with all reviewers that NAS is a very important field of study, that the experiments are interesting, and that purely empirical papers studying what works and what doesn't work (rather than introducing a new method) are definitely needed in the NAS community. But overall, for this particular paper, I agree with the 3 rejecting reviewers. The paper presents a lot of experiments, but I am missing novel deep insights or lasting overarching take-aways. The papers reads a bit like a log book of all the experiments the authors did, before having gone through the next iteration in the process to consolidate findings and gain lasting insight.

In a bit more detail, half the results in Section 4 use medium-sized super networks, which seem broken to me, yielding much worse performance than small super networks. I did not find any motivation for studying these medium-sized networks, no reason given for them to perform poorly, and none stating why the results are still interesting when the networks perform so poorly (apologies if I overlooked these). The poor performance may be due to using a training pipeline that works poorly for these larger networks, but this is hard to know exactly without further experiments. I would either try to fix these networks' performance or drop them from the paper entirely, as I do not see any insights that can be reliable gained from the current results. As is, I believe these results (accounting for half the plots in the paper) only muddy the water and are preventing a crisp presentation of insightful results.

Another factor that I find unfortunate about the paper is that it only uses NAS-Bench-201 for its empirical study, and even for that dataset, mostly only the CIFAR-10 part. After getting rid of isomorphic graphs from the original 15625 architectures, NAS-Bench-201 only has 6466 unique architectures (see Appendix A of NAS-Bench-201), while, e.g., NAS-Bench-101 has 423k unique architectures. As the authors indicate themselves in their section "Grains of Salt", it is unclear whether insights gained on the very small NAS-Bench-201 space generalize to larger spaces. I therefore believe that there should also be some experiments on another, larger space, to study how well some of the findings generalize. An additional benchmark that the authors could have directly used without performing additional experiments themselves is the NAS benchmark NAS-Bench-1shot1 (ICLR 2020: https://openreview.net/forum?id=SJx9ngStPH), which studies 3 different subsets of NAS-Bench-101, and which was created to allow one-shot methods to use the larger space of evaluated architectures in NAS-Bench-101.

Minor comments:
- It reads as if the authors performed 5 runs, computed averages of the outcomes, and then computed correlation coefficients. That would be a suboptimal experimental setup, though; in practical applications, only one run of the super network would be run, and therefore, in order to assess performance reliably, one should compute correlation coefficients for one run at a time, and then obtain a measurement of reliability of these correlation coefficients across the 5 runs.
- The y axis in Figure 2 appears to be broken: for example, in the left column it goes from 99.978 to 99.994, and the caption says these should be accuracy predictions of NAS-Bench201. However, even the best architectures in NAS-Bench201 only achieve around 95% accuracy.


Overall, I recommend rejection for the current version of the paper.
Going forward, I encourage the authors to continue this line of work and recommend that they iterate over their experiments and extract crisp insights from their experiments. I also recommend performing experiments with a much larger search space than that of NAS-Bench-201 to assess whether the findings generalize.